# Towards Understanding The Winner-Take-Most Behavior Of Neural Network Representations

## Abstract

Understanding the generalization ability of neural networks is a long-standing goal of the machine learning community. Despite sufficient representational complexity to memorize large data sets, modern neural networks are able to learn solutions to their optimization problem that generalize well to unseen data. In this paper, we explore how the neuron-level representations of the training samples in a data set can be analyzed to differentiate between networks that have learned to generalize and networks that merely memorize. For this purpose, we introduce a synthetic data set specifically crafted to allow for an easy comparison of how networks treat simple patterns. We show that comparing how training samples presenting different patterns are represented by neurons can provide key insights as to what differentiates memorized and generalized networks. We observe that the training process progressively increases the average pre-activation of the most activated patterns of a class and decreases the average pre-activation of the least activated patterns of said class in each neuron, a winner-take-most phenomenon. In order to solve the classification problem, the network seems to apply a divide-and-conquer strategy, where different neurons specialize in the classification of different patterns of a class. We also explore the effect of various parameters of our experimental configuration on these findings and describe three necessary conditions for it to appear. Finally, we provide an intuitive explanation of why this phenomenon occurs, drawing links with existing work on sample difficulty, coherent gradients, and implicit clustering.

## 1 Introduction

It is accepted knowledge at this point that despite SGD being a global, end-to-end optimization algorithm, neurons exhibit localized behavior during SGD-based training. Most notably, a large body of work has observed empirically that the hidden neurons (or feature maps) of trained deep neural networks capture interpretable features (Zeiler & Fergus, 2014; Simonyan et al., 2014; Yosinski et al., 2015; Zhou et al., 2016; Bau et al., 2017; Cammarata et al., 2020). Additionally, Carbonnelle & De Vleeschouwer (2021) showed that individual neurons play a crucial role in the presumed clustering mechanisms of deep learning.

In this paper, we attempt to describe causal implicit clustering mechanisms that lead to this behavior, by studying SGD from the perspective of hidden neurons. We make several simplifying assumptions on the data sets, models, and training process. These are detailed in Section 2. We begin by motivating this modus operandi in Section 3.1, by showing an interesting and counterintuitive result of network training in noisy environments, highlighting the importance of striking the right balance between training set signal and noise for a network to be able to generalize well. This leads us to consider the neural representations of various clusters of samples in models achieving diverse amounts of generalization. Our experiments of Section 4 on these representations reveal a winner-take-most behavior similar to that of several existing clustering algorithms (Martinetz & Schulten, 1991; Fritzke, 1997); more precisely, we show that in each neuron, the training process progressively increases the average pre-activation of the most activated clusters of a class, while decreasing the average pre-activation of the least activated clusters of said class. This can sometimes even lead to neurons differentiating clusters belonging to the same class more strongly than clusters in different

classes, as discussed in Section 4.1. The network thus seems to apply a divide-and-conquer strategy where different neurons specialize for the classification of different clusters in a class.

In Section 5, we provide an intuitive explanation of the possible origin of this phenomenon based on the Coherent Gradients Hypothesis of Chatterjee (2020) and work on the training dynamics w.r.t. sample difficulty by Arpit et al. (2017). We also support the generality of our observations by including an analysis of how, despite its apparent simplicity, our setup exhibits and provides insights on some of the phenomena occurring in state-of-the-art models, such as the benefits of implicit clustering abilities (Carbonnelle & De Vleeschouwer, 2021).

Finally, Section 6 summarizes our findings and provides some avenues potential avenues for further work

## 2 APPROACH

In order to make our results as interpretable as possible and to enable us to extract causal links between various aspects of the problem, we work in a curated environment, with many simplifying assumptions on the data, model, and training process that were used.

### 2.1 DATA SETS

A large part of our research was done using a synthetic data set crafted specifically to suit our particular needs, $\mathbb{S}$. The goal of this synthetic data set was to create an easily modifiable data set with hidden hierarchical patterns (also sometimes referred to as clusters) below the level of the supervision labels. The overview given here uses default values for variable parameters of the data set; when results were obtained with different settings, this is explicitly specified. This data set is composed of $n_{\text{data}} = 21000$ vectors of $n_{\text{dim}} = 500$ elements, such that $\boldsymbol{x} \in \mathbb{R}^{500}$, with 15000 of those being used for training ($\mathbb{S}_{\text{train}}$) and 6000 for testing ($\mathbb{S}_{\text{test}}$).

Each sample presents one of $n_{\text{pat}} = 30$ possible nonoverlapping patterns, defined as the sum of a binary pattern mask with $l_{\text{pat}} = 5$ nonzero entries set to $\delta = 1$, and additive Gaussian random noise with zero mean and $\sigma = 0.4$ standard deviation on each of its components. As such, patterns form clusters in the sample space, with the centroids being the vectors corresponding to the noiseless pattern masks. The patterns are then aggregated into two possible supervision labels, with $n_{\text{pat}}/2 = 15$ patterns per class label. Importantly, this means we never explicitly give models access to pattern labels. The only information they receive is the class label. By default, all patterns are equally represented in the data set. Algorithm 1 further details the data-generating procedure.

---

**Algorithm 1** Generation procedure for the synthetic data set $\mathbb{S}$

---

**Input:** $n_{\text{dim}}, l_{\text{pat}}, n_{\text{pat}}, n_{\text{data}}, \delta, \sigma$ {Number of dimensions, pattern length, number of patterns, number of samples, pattern separation, and noise standard deviation.}

Initialize $n_{\text{data}}$ normally distributed samples $\boldsymbol{x}^{(i)}$, $i = 1, \ldots, n_{\text{data}}$, such that $x_j^{(i)} \sim \mathcal{N}(0, \sigma)$ for $j = 1, \ldots, n_{\text{dim}}$.

Assign a pattern label $p^{(i)}$ to each sample $\boldsymbol{x}^{(i)}$, with domain $\{1, \ldots, n_{\text{pat}}\}$. By default, all patterns appear the same number of time in the data ($n_{\text{data}}/n_{\text{pat}}$ times).

Assign a binary mask $\mathcal{P}_k \subseteq \{1, \ldots, n_{\text{dim}}\}$, $|\mathcal{P}_k| = l_{\text{pat}}$, for each pattern label $k$. If no overlap is desired, choose $\mathcal{P}_k \subseteq \left(\{1, \ldots, n_{\text{dim}}\}\right) \setminus \bigcup_{1 \leq j < k} \mathcal{P}_j$ instead. The pattern centroids are then $c_k = \sum_{i \in \mathcal{P}_k} \delta \boldsymbol{e}_i$ in the canonical basis.

Assign class labels, aggregating possible pattern labels into two classes: sample $\boldsymbol{x}^{(i)}$ has label $\mathcal{C}^{(i)} := \lfloor p^{(i)}/(n_{\text{pat}}/2 + 1) \rfloor$.

**for** $i \leftarrow 1, \ldots, n_{\text{data}}$ **do**
    **for** $j \in \mathcal{P}_i$ **do**
        $\boldsymbol{x}^{(i)} \leftarrow \boldsymbol{x}^{(i)} + \delta \boldsymbol{e}_j$ {Add separation to pattern components.}
    **end for**
    $\boldsymbol{x}^{(i)} \leftarrow \boldsymbol{x}^{(i)}/\|\boldsymbol{x}^{(i)}\|$ {$L^2$-normalize.}
**end for**

---

In order to show applicability to broader settings, we reproduced some of our results on a modified version of the MNIST data set (LeCun, 1998) obtained by aggregating the digits into two classes of 5 digits each which are used as supervision labels ($\{0,\ldots,4\}, \{5,\ldots,9\}$); we call this data set Aggregated MNIST. By analogy with the synthetic data set, each digit corresponds to a distinct "pattern." This is only an approximation, as digits themselves present many "intraclass clusters" as discussed by Carbonnelle & De Vleeschouwer (2021), but it is sufficient for our purposes.

## 2.2 MODELS AND TRAINING PROCESS

We want our models to be as interpretable as possible, and for that reason, use multilayer perceptron (MLP) networks (Rosenblatt, 1958). More specifically, we train MLP networks with a single hidden layer. The hidden layer is composed of 1000 neurons without biases. The output layer is composed of one neuron with a sigmoid activation function. Using the dimensions of the data set in Section 2.1, the model $\mathscr{M}$ thus simply computes

$$f(\boldsymbol{x}) = \sigma\big(\boldsymbol{W}_2^\top \rho(\boldsymbol{W}_1^\top \boldsymbol{x}) + \boldsymbol{b}_2\big), \tag{1}$$

where $\boldsymbol{W}_1 \in \mathbb{R}^{500 \times 1000}$ is the weight matrix between the input and hidden layers, $\boldsymbol{W}_2 \in \mathbb{R}^{1000 \times 1}$ is the weight matrix between the hidden and output layers. $\boldsymbol{b}_2$ contains the biases between the hidden and output layers. Finally, $\sigma$ and $\rho$ denote the sigmoid and ReLU (Fukushima, 1975) activation functions, respectively.

To train the models in Section 3.1, we use stochastic gradient descent (Amari, 1993) with a binary cross-entropy loss, using a learning rate of $1$ and a batch size of $64$. We train models until they reach perfect accuracy on the training set, i.e. they correctly classify all training samples. For all other parameters, the default implementations provided by the PyTorch (Paszke et al., 2019) module were used.

For the MLP models trained in Section 4, we modify our training process slightly by using the Layca optimizer (Carbonnelle & De Vleeschouwer, 2019). We train in full-batch mode for the synthetic data set and with batch size 20000 for Aggregated MNIST, to avoid the sampling noise inherent to small-batch training (while Keskar et al. (2017) argued the importance of small batch sizes for generalization, this view has now been relativized by several others (Hoffer et al., 2017; Goyal et al., 2017; You et al., 2017; Geiping et al., 2022)). We train for a fixed number of $400$ epochs on the synthetic data set and $100$ on Aggregated MNIST, and with a learning rate of $3^{-3}$, which we reduce by a factor of $5$ at epochs $375$ and $85$ respectively. Models trained on Aggregated MNIST also use batch normalization (Ioffe & Szegedy, 2015). These models reach test accuracies of $96.1\,\%$ and $98\,\%$ for $\mathbb{S}$ and Aggregated MNIST, respectively.

## 3 WHAT NOISE TELLS US ABOUT GENERALIZATION

In this section, we attempt to determine how the performance of a network depends on the amount of noise present both in the training set and test set.

## 3.1 TRAINING AND TEST SET NOISE

We begin our exploration by considering how network performance is affected by the amount of noise in the data. Figure 1 shows what happens to the performance of models when the standard deviations of the training and test set noise are changed ($\sigma_{\text{train}}$ and $\sigma_{\text{test}}$, respectively). We see that networks trained with a moderate amount of noise ($\sigma_{\text{train}} \approx 0.4$) tend to perform better than models trained on a more or less noisy training set, and this for any value of the test set noise standard deviation $\sigma_{\text{test}}$.

We attribute this to the fact that good generalization in this situation relies on the network's ability to perform two sub-tasks well: (i) it needs to be able to separate classes (and thus patterns) correctly (ii) while being robust to the noise in the data. The first of these tasks requires the training set to be sufficiently "systematic," i.e. on average, the patterns should explain most of the difference between two samples; when the amount of added noise is so great that it obscures the pattern separation, a network will instead tend to memorize large parts of its training set, resulting in poor generalization performance. The ability of a network to memorize its data set was shown in Arpit et al. (2017).

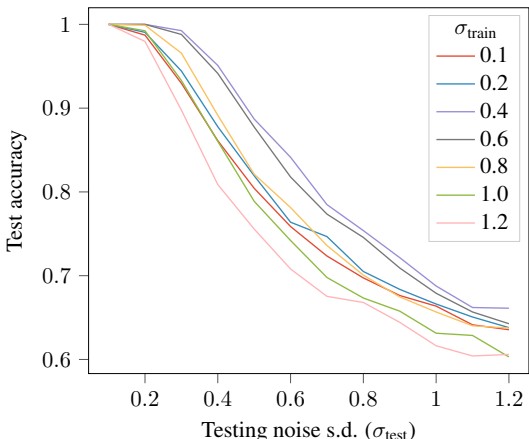

Figure 1: Test accuracies when varying $\sigma_{\text{train}}$ and $\sigma_{\text{test}}$. Models trained with a moderate amount of training noise ($\sigma_{\text{train}} \approx 0.4$) outperform other models over a wide range of test noise values.

This is what we observe on Figure 1 for the models trained with higher values of $\sigma_{\text{train}}$: they do not perform well, even in situations where they should have the advantage of domain specificity, i.e. when training and testing noise is similarly high.

The second task requires the network to see sufficient amounts of noise during its training process. Models trained with low values of $\sigma_{\text{train}}$ do not learn to handle noise, since their training situation does not confront them with many noisy samples. On Figure 1, this can be seen by the models performing well when $\sigma_{\text{test}}$ is low (because they are unlikely to deal with many significantly noisy test samples then), but quickly falling off when test set systematicity is reduced. The network "sees" the patterns when they are abundantly clear, but is unable to do so in the presence of any meaningful amount of noise.

## 3.2 Internal Representations for Generalization

Considering the observations made on Figure 1, we now turn our attention towards the neural representations of samples in our networks. Going up one semantic level above individual samples, we look at the average pre-activations in the hidden layer neuron with the highest absolute connection to the output neuron[1] for all test samples (for $\sigma_{\text{test}} = 0.4$) presenting a given pattern. More precisely, we are interested in how these average pre-activations evolve throughout the training process. Figure 2 compares what this evolution looks like in a model that generalizes well (Figure 2a, with 95.1 % test accuracy) and a model that does not (Figure 2b, with 80.9 % test accuracy). The generalizing model, which was trained with $\sigma_{\text{train}} = 0.4$, has a qualitatively different way of treating patterns than the model trained with $\sigma_{\text{train}} = 1.2$. In Section 4, we explore why this is the case.

## 4 Hierarchical Patterns as Drivers of Implicit Clustering

In this section, we attempt to find out what makes the neuronal representations of generalizing models so particular, and we draw links between the phenomena we observe and prior work on neural network generalization properties.

## 4.1 A Winner-Take-Most Mechanism

We record throughout the duration of the network training process, two neuron-level training signals for a hidden neuron $n^{(i)}$, for each sample $\boldsymbol{x}^{(j)}$, after each epoch $k$: (i) the pre-activation $z_{i;k}^{(j)} = (\boldsymbol{W}_{1;k}^{\top})_{i,:}\boldsymbol{x}^{(j)}$, and (ii) the partial derivative of the loss w.r.t. the activation, $(\partial L/\partial a_i^{(j)})_k$,

---

[1]This choice of neuron is for clarity and illustrative purposes, as it has the most impact on the prediction of the network; the qualitative behavior we describe can be observed in most neurons of the network.

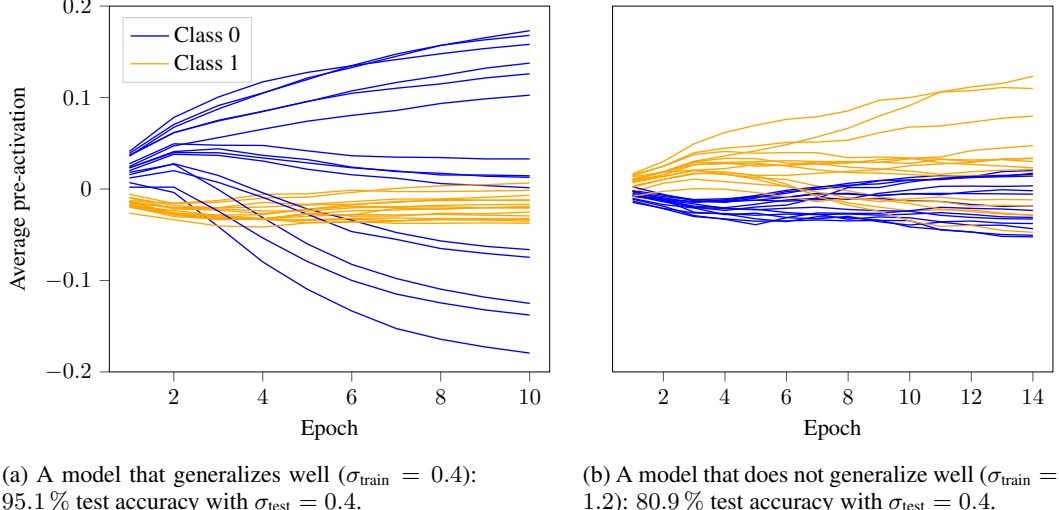

(a) A model that generalizes well ($\sigma_{\text{train}} = 0.4$): 95.1 % test accuracy with $\sigma_{\text{test}} = 0.4$.

(b) A model that does not generalize well ($\sigma_{\text{train}} = 1.2$): 80.9 % test accuracy with $\sigma_{\text{test}} = 0.4$.

Figure 2: Neural representations as defined by average pattern pre-activation throughout training. The model on the left generalizes better than the model on the right, and the representations in its neurons are qualitatively different.

where $a_i^{(j)} = \rho(z_i^{(j)})$. Missing terms are defined in equation 1. After training, we again select the neurons with the strongest influence on the model's predictions, i.e. with the largest absolute weights in the output layer, to simplify exposition, as in Section 3.2. These results are shown in Figure 3.

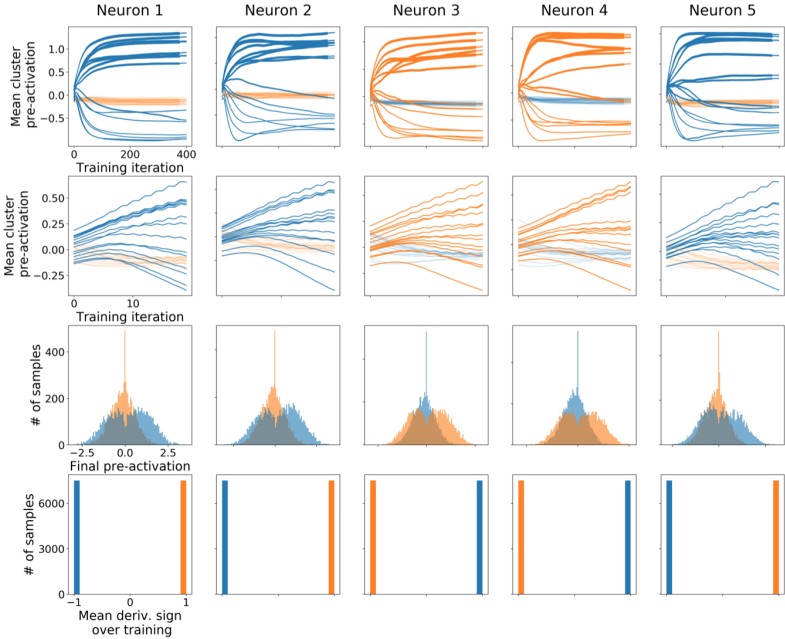

Figure 3: Evolution of pre-activations and loss derivatives on the synthetic data set. The first two rows represent the evolution of each pattern's average pre-activation during training. The third row displays the resulting pre-activation distributions at the end of training. The fourth row displays a histogram of the sign of the derivative of the loss w.r.t. the neuron activation, averaged over all training steps.

We observe that during training, each neuron consistently differentiates the patterns of a class according to a winner-take-most mechanism: patterns with larger average pre-activations are pushed towards even larger pre-activations, while patterns with smaller pre-activations are pushed towards even smaller pre-activations. Perhaps most surprising is the observation that from the perspective of a single neuron, this unsupervised mechanism can be more impactful than the supervised learning process: neurons sometimes differentiate patterns belonging to the same class more strongly than patterns from different classes.

A second observation concerns the partial derivative of the loss w.r.t. the activation. More precisely, we compute, for each sample, the mean sign of this derivative across all steps of the training process. This allows us to determine whether an increased activation generally benefits or penalizes the classification of a given example, resp. when the average sign is negative or positive. This sign appears to be correlated with the sample's class across the whole training process: the samples of the negative mean derivative sign class should always be pushed towards larger activations and those of the positive mean derivative sign class to smaller activations. However, in practice, we observe the winner-take-most mechanism described previously occurring with samples of the negative modal derivative sign class. In Section 4.2, we explore why some of the clusters of this class are pushed towards smaller activations, despite being associated with negative derivatives.

## 4.2 TOWARDS UNDERSTANDING THE MECHANISM

To better understand the emergence of this winner-take-most phenomenon, we perform an ablation study that identifies the necessary ingredients for it to occur. We then provide an intuitive framework to explain the phenomenon based on properties of "difficult" training samples, gradient coherence in ReLU neurons, and the divide-and-conquer approach of the hidden neurons.

### 4.2.1 AN ABLATION STUDY

Figure 4 shows that despite its apparent simplicity, the setup of Section 2 contains several elements without which the winner-take-most phenomenon does not occur: multiple hidden neurons, sufficient amounts of noise (which we already discussed in Section 3.2), and nonlinearity in the activation function. Without these necessary conditions, hidden neurons behave like the output neuron, classi-

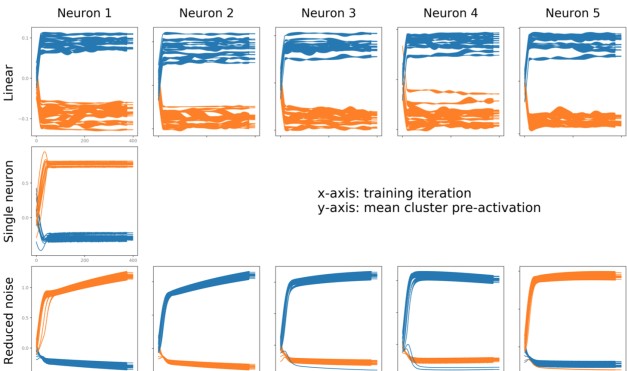

Figure 4: Several conditions are necessary for the winner-take-most phenomenon to appear. Average pre-activation of each cluster across training for a model without ReLU activation layer (first row), with a single hidden neuron (second row), or trained on a less noisy data set (third row).

fying data according to the two supervision classes, without taking intraclass clusters into account; we also observe a decrease in model performance, with test accuracies ranging from $80\,\%$ to $84\,\%$. Throughout the rest of this section, we explore how these necessary conditions lead us to certain intuitions about neural network training.

### 4.2.2 ON THE ROLE OF DIFFICULT TRAINING SAMPLES

The second row of Figure 3 shows that around the 6th epoch of the training process, we observe some clusters being "pushed" in the same direction as clusters of the opposite class. These are

the "losing" clusters of the class subject to the winner-take-most mechanism. Initially, this local behavior might seem contrary to the global objective: differentiating examples from their opposite class. In particular, the fourth row shows that derivatives associated with these clusters are negative, aiming in the opposite direction they actually go.

This apparent contradiction might be explained by considering the role of difficult training samples: samples that are hard to classify because the additive noise decreases their correlation with their actual class and increases the correlation with the other class. As such, their contribution to the total loss progressively increases throughout training, as more of the "regular" training samples are correctly classified, and eventually, they become the dominant force driving the gradient directions. When this happens, regular samples get pushed in a direction opposed to their associated gradient.

These exact dynamics can be observed in Figure 5. In order to formally define what constitutes a

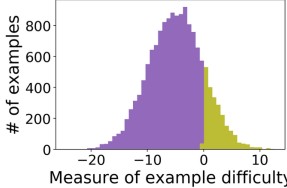 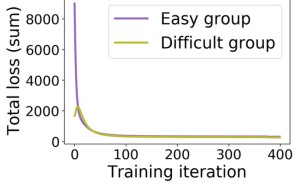

Figure 5: The role of difficult samples. (Left.) The distribution of the difficulty coefficient. (Right.) The evolution of the easy and difficult groups' share in the total loss during training.

difficult sample, we compute a difficulty coefficient $d(\boldsymbol{x}^{(i)})$ for a sample $\boldsymbol{x}^{(i)}$ with class label $\mathcal{C}^{(i)}$, defined as

$$d(\boldsymbol{x}^{(i)}) = \left\langle \boldsymbol{x}^{(i)}, \sum_{k \notin \mathcal{C}^{(i)}} c_k \right\rangle - \left\langle \boldsymbol{x}^{(i)}, \sum_{k \in \mathcal{C}^{(i)}} c_k \right\rangle. \tag{2}$$

This coefficient compares the sample to the combined centroids of its class and of the other class. Difficult samples are those with a positive difficulty coefficient. We observe that the loss associated with these difficult samples grows in the initial stage of training, meaning contradictory gradients will appear. Very quickly (around the 6th epoch), the share of the difficult group matches and even exceeds the share of the regular group; this happens around the same time the winner-take-most phenomenon appears in Figure 3. These observations align with the conclusion one can draw from Section 3.2 or the ablation study with reduced noise of Section 4.2.1: when noise is reduced, the share of difficult samples decreases, preventing the winner-take-most mechanism from occurring.

### 4.2.3 ON THE ROLE OF ReLU

As the overall gradient is computed as a sum of per-sample gradients, the directions that are coherent across multiple training samples are reinforced (Chatterjee, 2020; Zielinski et al., 2020). As the derivative of the ReLU activation function is $0$ for negative values, training samples that do not activate a given neuron do not contribute to the gradient associated with that neuron's weights. As such, for samples presenting a certain pattern, only those that activate the neuron reinforce each other.

In our simple single hidden layer MLP, the samples associated with each cluster share a common pattern in the input signal (by virtue of their definition in the data set of Section 2.1), and a common pattern in the back-propagated error signal (because their supervision class is the same). The number of samples that activate the neuron affects the relative share of each cluster in the gradient associated with the weights of the neuron. In particular, the average pre-activation of a cluster in a neuron is proportional to the share of that cluster in the weight gradient of the neuron. This could explain why difficult training samples impact clusters with smaller average pre-activations more strongly, as discussed in Section 4.2.2, causing them to "lose" the competition. On the contrary, clusters with larger average pre-activations will be less affected by difficult training samples, allowing them to "win" the competition. This idea that ReLU layers are key for differentiating "winning" clusters from "losing" ones is also coherent with the ablation study of Section 4.2.1, which shows that the winner-take-most mechanism does not occur in models with linear activation layers, i.e. without ReLU.

### 4.2.4 A DIVIDE-AND-CONQUER STRATEGY

Our ablation study of Section 4.2.1 demonstrates that a neural network with a single hidden neuron does not exhibit a winner-take-most mechanism. Intuitively, this can be explained as follows: the winner-take-most mechanism leads to local misclassification of multiple clusters; this is only viable if the local misclassification is counterbalanced by a correct classification in other neurons. A divide-and-conquer approach, where different neurons focus on the classification of different clusters, is then perfectly compatible with the winner-take-most phenomenon. The neuron-level differences in favored clusters are determined by the random weight initialization, which impacts the initial average pre-activation of each cluster.

### 4.2.5 WHY DOES THE MECHANISM AFFECT A SINGLE CLASS?

Jointly considering the role of difficult training samples, gradient coherence, and divide-and-conquer strategies also helps explain why the winner-take-most mechanism only applies to samples of the class associated with negative derivatives, i.e. whose activations should increase during training. Indeed, for this class, pushing a cluster in its "opposite" direction simultaneously leads to a deactivation of some of its samples and a reduction in this cluster's share of the neuron's gradient. This further promotes the correct classification of the associated difficult training samples. On the contrary, pushing clusters of the class associated with positive derivatives in their opposite direction increases the number of samples that activate the neuron (and the share of the cluster in the neuron's gradient), promoting the correct classification of these clusters in this specific neuron.

## 5 RELATED WORK IN STANDARD DEEP LEARNING SETTINGS

Our work reveals the emergence of a winner-take-most mechanism and provides intuitions and experiments to understand it. However, these contributions remain limited to relatively simple and shallow neural network architectures trained on a synthetic data set. In order to support the generality of our results and to show that our empirical observations and intuitions still hold in standard deep learning settings, we first discuss several works that studied difficult training examples and gradient coherence in standard deep learning settings, highlighting the connections with the intuitions described in Sections 4.2.2 and 4.2.3. Second, we demonstrate that our simple setup exhibits several phenomena occurring in standard settings and provide empirical evidence that these phenomena are reminiscent of winner-take-most mechanisms.

### 5.1 LINK WITH TRAINING DYNAMICS W.R.T. SAMPLE DIFFICULTY

Relating different notions of sample difficulty to the speed at which samples are learned has been attempted many times. Arpit et al. (2017) showed across different initializations and permutations of the training data that many samples are consistently (mis-)classified after a single epoch of training, which led them to conjecture that "deep learning learns simple patterns first, before memorizing." Mangalam & Prabhu (2019) showed empirical evidence that neural networks tend to learn "shallow-learnable" samples first, i.e. samples that are correctly classified by non-deep learning approaches. Jiang et al. (2021) characterized samples by their consistency score, as defined by their expected accuracy as a held-out sample given training sets of varying size. Figure 10 of this paper displays the training curves associated with the training samples, grouped by consistency score. It reveals that samples with higher scores are learned before those with lower scores. While this aspect is not discussed in the original paper, Figure 10 also reveals that the accuracy of examples with a low consistency score decreases in the first epochs of training. This suggests that the gradients of low-scoring examples are "contradictory" to the ones of high-scoring examples.

In Section 4.2.2, we defined the difficulty of training samples in equation 2 by their correlation with the opposite class relative to their correlation with their true class. In accordance with the observations conducted in standard settings, we observe in Figure 5 that in our simple setup, (i) easy samples are learned before difficult ones and (ii) the loss of difficult samples increases in the first training iterations, suggesting the presence of contradictory gradients.

## 5.2 LINK WITH THE COHERENT GRADIENT HYPOTHESIS

Chatterjee (2020) recently introduced the Coherent Gradient Hypothesis, which states that gradient coherence plays a crucial role in the generalization abilities of deep neural networks. Zielinski et al. (2020) provides multiple experiments to support this hypothesis in the context of standard deep learning settings involving the ImageNet dataset and ResNet models with 18 layers. These works justify the role of gradient coherence with the following intuition: because gradients are the sum of per-sample gradients, coherence is stronger in the directions where the per-sample gradients are more similar. The changes to the network during training are thus biased towards those that simultaneously benefit many samples. They further argue that such a bias is beneficial for generalization, based on algorithmic stability theory. However, the previous intuition only holds at the very beginning of training, when most samples are misclassified by the model. As we showed in Section 4.2.2, a small set of difficult training examples strongly influences the overall gradient once regular examples are correctly classified. In (Chatterjee & Zielinski, 2020; 2022), they provide a more extensive analysis of the evolution of gradient coherence during training. They conclude that their experiments provide additional evidence for the connection between the alignment of per-sample gradients and generalization, but that their data shows this connection is complicated. We believe that the winner-take-most mechanisms disclosed by our work and the intuitions described in Section 4.2.3 offer a promising path towards a better understanding of the relationship between gradient coherence and generalization.

## 5.3 LINK WITH THE BENEFITS OF IMPLICIT CLUSTERING ABILITIES

Carbonnelle & De Vleeschouwer (2021) proposed 5 measures of intraclass clustering that correlate with generalization in standard deep learning settings. These correlations occur across variations of 8 standard hyperparameters, such as data augmentation, depth, and learning rate. Two measures in particular, ($c_1$ and $c_3$), are applied at the neuron level, capturing the extent by which samples or subclasses from the same class are differentiated in a neuron's pre-activations.

These measures are thus closely related to the winner-take-most mechanism we studied in our simple setup, as it leads to the differentiation of clusters from the same class in a neuron's pre-activations. The observations are consistent with the experiments of their paper: the better the clusters of a class are differentiated, the better the generalization performance. Studying both implicit clustering abilities and mechanisms thus constitutes a coherent framework supporting the crucial role of implicit clustering in deep learning systems.

## 6 DISCUSSION AND FUTURE WORK

In this work, we have designed a synthetic data set with the explicit goal of looking at how patterns or clusters in the data can be used as a higher-level semantic abstraction to help us understand the workings of neural networks. Through experiments with noisy data sets, we found that in our simplified setup, an optimal amount of training noise exists for generalization. We also observed that networks that generalize well (such as those obtained in that way) tend to learn remarkable neuronal representations that form as a consequence of a winner-take-most-mechanism. We investigated what this mechanism looks like, and found some necessary conditions for its appearance: sufficient noise, nonlinearity, and representational complexity. Through the lenses of sample difficulty and gradient coherence, we provided an intuitive explanation for the phenomenon we observed. We also looked at how it relates to the benefits of implicit clustering ability in deep neural networks and found that they are related through the differentiation of clusters from the same class.

Future work on this topic should further dig into the links between the winner-take-most mechanism and observations in standard deep learning settings. One such example would be to look at the regularizing effects of data augmentation and pre-training and how they relate to the winner-take-most phenomenon since one can hypothesize that these would both affect the number of "difficult" samples being provided to the network. Similarly, studying the effect of depth could help bridge the gap with real-world neural networks, while also exploring how depth fairly consistently leads to good generalization. Finally, the effect of large learning rates is likely to also interplay with the winner-take-most phenomenon, and a better understanding of how they relate to each other could help shine light on empirical observations with the use of large learning rates.

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

# A  APPENDIX

## A.1  THE WINNER-TAKE-MOST MECHANISM ON AGGREGATED MNIST

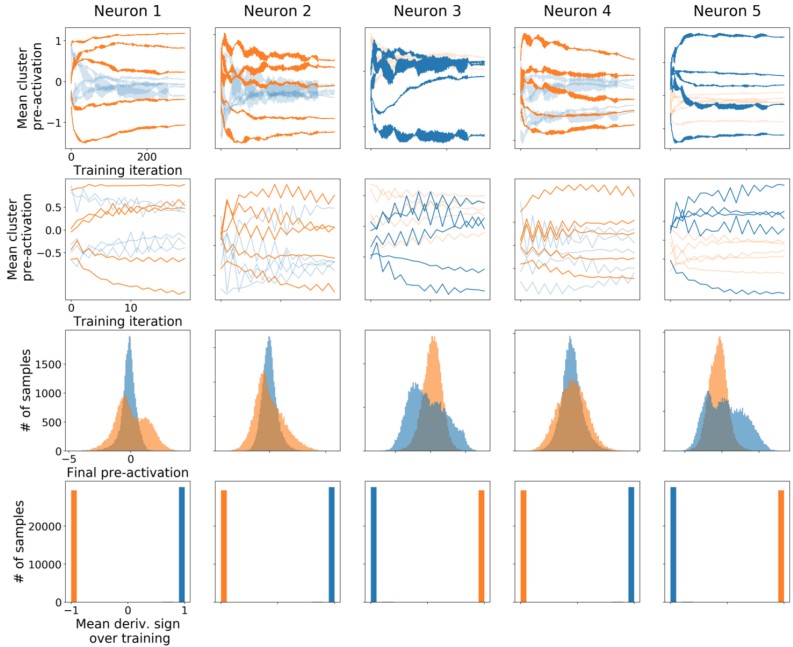

Figure 6: Evolution of pre-activations and loss derivatives on the Aggregated MNIST data set. The first two rows represent the evolution of each pattern's average pre-activation during training. The third row displays the resulting pre-activation distributions at the end of training. The fourth row displays a histogram of the sign of the derivative of the loss w.r.t. the neuron activation, averaged over all training steps.

