# OpenReview forum: "Towards Understanding The Winner-Take-Most Behavior Of Neural Network Representations"
_ICLR.cc/2024/Conference — ICLR 2024 Conference Withdrawn Submission_

### Official Review · Reviewer_xKTf · 2023-10-13

**Soundness:** 2 fair
**Presentation:** 2 fair
**Contribution:** 2 fair
**Rating:** 3
**Confidence:** 3

**Summary:**

This paper studies a winner-take-most mechanisms in two-layer neural networks for a synthetic dataset S and aggregated MNIST, i.e., a neuron becomes more active (in terms of preactivtion) for one class, but becomes less active for the other class. They also show that neural networks seem to apply a divide-and-conquer strategy, where each neuron focuses on different patterns of the data. I believe the overall story makes sense, but the arguments are not sufficiently supported by empirical evidence.

**Strengths:**

The overall structure is clear with exhaustive literature review.

**Weaknesses:**

Most empirical findings in this paper are qualitative (which is fine and good for illustrating your idea). To make big claims (winner-take-most, divide-and-conquer), quantitative metrics are needed to evaluate these effects. Although I feel obligated to suggest larger-scale experiments, I feel even the current setups are not fully studied yet. I'm happy to change my score if more in-depth experiments are carried out on your current datasets. See my questions below.

**Questions:**

This paper centers around a key idea "winner-take-most"
* Although I roughly get the idea and think it does make sense, its meaning seems reloaded multiple times. E.g., in Section 4.1 "a winner-take-most mechanism: patterns with larger average pre-activations are pushed towards even larger pre-activations, while patterns with smaller pre-activations are pushed towards even smaller pre-activations." and in Section 4.2, "A divideand-conquer approach, where different neurons focus on the classification of different clusters, is then perfectly compatible with the winner-take-most phenomenon.". Are the two definitions equivalent? Is divide-and-conquer just a paraphrase of winner-take-most, or is one contained in the other?
* The motivation of studying the winner-take-most mechanism is not clear and why it matters, especially given its subtlety (Section 4.2.1-4.2.3).
* Section 4.2.4/4.2.5 read interesting, but feel incomplete. More experiments need to be carried out to characterize the phenomenon. For example, how to define a metric to quantify divide-and-conquer?

Minor issues:
* At the end of introduction, missing a period.
* learning rate 3^{-3}, you mean 10^{-3}?

---

### Official Review · Reviewer_s4ij · 2023-10-29

**Soundness:** 2 fair
**Presentation:** 3 good
**Contribution:** 2 fair
**Rating:** 3
**Confidence:** 3

**Summary:**

This paper delves into the "winner-take-most" dynamic observed in the representations learned by neural networks within binary classification tasks. It discusses how in such scenarios, one class tends to be represented with higher activation magnitudes, while the other is characterized by comparatively lower activations. The paper constructs synthetic data where the data points are the sum of binary masks with noise ($\sigma_\text{train}$). A pattern is a sum of 5 distinct binary masks and both classes contain an equal number of distinct and non-overlapping patterns. The model studies is a two-layer neural network that is trained with SGD on this synthetic dataset (setting 1) and also an MLP model trained with different hyperparameters on the MNIST dataset (setting 2).

In setting 1, the paper finds that the amount of noise in the training set has an important effect on the generalization performance of the model, and a moderate amount of $\sigma_\text{train}$ leads to the best generalization performance. They discovered that usually, the activation of one class's patterns will have large activations whereas the activation of the other class' patterns would have small activation (near 0). The paper then studies why this phenomenon occurs and identifies that multiple hidden units, non-linearity, and noise are needed for this "winner-take-most" effect to occur.

The paper then conjectures that the reason why this phenomenon occurs is due to the fact that there are "difficult" examples that are close to the other class due to noise. The representation of these difficult examples is first pushed away from the representation of the other examples of the same class (evidenced by the initially increased loss) and then pushed away from the representation of the other class.

Finally, the paper explains the potential role of non-linearity and multiple neurons. The paper also discusses the relationship of the observed phenomena to other phenomena in the literature.

**Strengths:**

- The paper is fairly well-written and proposes several fairly interesting explanations for empirical phenomena.
- The experimental setup of the paper is carefully designed and well-explained and the conclusions in the paper are presented fairly clearly. They are able to highlight some potential reasons behind the observed phenomena.

**Weaknesses:**

While I find the experiments and proposals interesting, the main concern I have for the paper is that it is not clear to me how much of these conclusions carry to the real data and real architecture. I would be happy to increase my rating if the authors can show more evidence that the claims are really relevant for real models.

- The study primarily relies on a synthetic dataset, raising questions about the applicability of its findings to real-world data and architectures. Experiments on the MNIST dataset already show behaviors that differ from the synthetic data, suggesting that the observations may not be generalizable. For example, I find it odd that the authors chose the Layca optimizer which is not a widely used optimizer. Further experimentation with different optimizers, such as SGD or Adam, particularly on more complex datasets like CIFAR10 (which wouldn't be too hard to run), could provide a clearer indication of the robustness of the conclusions.
- The research is limited to binary classification. It is unclear how the findings would translate to multiclass problems where the activation dynamics are inherently more complex due to the difficulties of mapping all non-target classes to low magnitudes.
- The paper’s definition of example difficulty is exclusively tied to the model's initialization rather than the characteristics of the data itself, which seems to oversimplify real-world data complexity.
- The link between the "winner-take-most" phenomenon and model generalization is not convincingly established. Without stronger empirical evidence, it is difficult to determine whether this is a genuine factor in generalization or merely a byproduct of other underlying factors. More empirical evidence could help clarify this relationship.

Overall, while the paper presents interesting initial findings, its conclusions would be significantly strengthened by additional experiments that consider more realistic datasets, optimizers, and classification scenarios, as well as a more nuanced consideration of example difficulty and its implications for model generalization.

**Questions:**

- What happens if you use a more realistic architecture (e.g., ResNet)? It seems to me that all of your analysis would easily be applicable to different architectures as long as it remains a binary classification. This would greatly strengthen my trust in the proposed mechanisms if they still hold.
- can you apply this reasoning to non-last-layer activations? How does that affect the conclusion of your observations? How does your framework account for the effects of different architecture or features at different depths?
- For the noise claim, can you show that it holds for real datasets and real architecture?
- Similarly, for the non-linearity, can you show that it holds for real datasets and real architecture? What if you do not use ReLU but other activations?
- I don't quite understand the explanation for "Why does the mechanism affect a single class?". This in my opinion is one of the most important observations in this paper. Can you provide a more detailed explanation and perhaps some illustrations?

---

### Official Review · Reviewer_u4t5 · 2023-11-05

**Soundness:** 1 poor
**Presentation:** 2 fair
**Contribution:** 1 poor
**Rating:** 1
**Confidence:** 3

**Summary:**

The paper aims to study the winner-take-most behavior of binary classification NNs by using a two-layer MLP and (mainly) a synthetic dataset with Gaussian noises (15 clusters per class) as well as the aggregated MNIST dataset. The paper presents preliminary results on the effect of training noises (Fig 1, 2), the emergence of the winner-take-most behavior (Fig 3, 6) and conditions when it disappears (Fig 4), then discusses the role of training difficulties (noises, Fig 5), ReLU, and number of hidden neurons on the winner-take-most behavior (Sec 4.2), finally cites previous works to support the generality of this paper (Sec 5).

**Strengths:**

+ [Clarity] The paper is overall easy to follow, although can be written more concisely.

**Weaknesses:**

- [Originality] The winner-take-all property has been widely used in previous works such as NN-based clustering algorithms [1] and it’s unclear how this paper contributes novelly to the understanding of this behavior with its extremely simplified settings, especially since most of the findings have been reported in previous works (Sec 5).
- [Quality] The quality of the paper is unacceptable due to the following issues:
1) The experimental setup is highly insufficient for a top-tier conference like ICLR, with overly simplified network, datasets and analyses (only scalar plots from single neurons instead of e.g. cluster analysis and/or visualization), leaving the results highly inconclusive.
2) The observed winner-take-most (divide-and-conquer) behavior could be simply due to insufficient training as it contradicts the neural collapse theory [2] which predicts the opposite, the collapse of all intraclass clusters, leaving the main results highly questionable.
3) Claiming that training noises are required for generalization on the synthetic dataset (Sec 3.1) is quite problematic, since sufficient training should generally lead to the max-margin solution (which generalizes) even without training noises [3]. The extremely large learning rate (1.0) could be causing the problem.
- [Significance] Given the critical issues in the paper’s originality and quality as stated above, it’s unfortunately hard to conclude that this work is significant or sufficiently promising.

[1] Clustering: A neural network approach, Neural networks, 2010.\
[2] Prevalence of neural collapse during the terminal phase of deep learning training, PNAS, 2020.\
[3] The Implicit Bias of Gradient Descent on Separable Data, JMLR, 2018.

**Questions:**

Please address the weaknesses as much as possible.

---

### Official Review · Reviewer_hMQT · 2023-11-08

**Soundness:** 3 good
**Presentation:** 3 good
**Contribution:** 2 fair
**Rating:** 6
**Confidence:** 4

**Summary:**

The authors study the generalisation abilities of moderns neural network. To this end, they carry out analyses on model representations  of training data to differentiate between networks that have learned to generalise and networks that merely memorise. In an analysis of a small network, they discover a winner-takes most behaviour, where average pre-activation of the most activated patterns of a class increase and the average pre-activation of the least activated patterns decrease. Further, they find that a divide-and-conquer strategy, different neurons specialising on different class patterns in a classification problem.

**Strengths:**

The paper addresses an important problem in machine learning interpretability, representation learning/analysis and model behaviour to study generalisability. The methodology presents a clear and detailed analysis of representations on a toy classification dataset and comes to sound conclusions (winner-takes-most, divide and conquer).

It is well written and good to follow.

**Weaknesses:**

Despite the soundness of the approach and detailed analyses, I am unsure about the novelty of this approach, as I find that related research is not sufficiently addressed. The NLP community has been producing a substantial body of work, and on different parts of (mostly transformer language) models: Some authors include Geva [1] (mostly on FF-Layers), Elena Voita (Attention mechanism) [2], follow-up work from David Bau, whom the authors cite with an older work (e.g., interpretability and model editing of factual knowledge in neurons [3]), and Anthropic's mechanistic interpretability [4]. As this research is on neuron representations of (textual) data in large language models I find it very important in the context of this work, and partially reminds me of winner-takes-most; except for an older paper from David Bau 2017, however, this is entirely missing from related work. Accordingly, I would encourage the authors to carefully study recent related work and also compare their work.
Furthermore, the approach only studies a small network on a toy dataset, although it is carried out thoroughly. A quantitative evaluation of this work on real-world data from some of the works pointed out above could be one option to address this issue.

[1] https://scholar.google.com/citations?user=GxpQbSkAAAAJ&hl=en
[2] https://scholar.google.com/citations?user=EcN9o7kAAAAJ&hl=th
[3] https://transformer-circuits.pub/2021/framework/index.html
[4] https://arxiv.org/pdf/2202.05262.pdf

**Questions:**

How would a comparison, and evaluation in comparison to existing works a language models look like? How does your work relate to research in NLP?

How could your approach relate to and help in model pruning, which removes not useful and unused neurons (see, e.g., [1]), and also knowledge distillation methods [2]?

[1] https://aclanthology.org/P19-1580/
[2] https://aclanthology.org/2023.acl-long.818